# Restoring Genetic Resource through In Vitro Culturing Testicular Cells from the Cryo-Preserved Tissue of the American Shad (*Alosa sapidissima*)

**DOI:** 10.3390/biology11050790

**Published:** 2022-05-22

**Authors:** Hong-Yan Xu, Xiao-You Hong, Chao-Yue Zhong, Xu-Ling Wu, Xin-Ping Zhu

**Affiliations:** 1Key Laboratory of Freshwater Fish Reproduction and Development (Ministry of Education), Key Laboratory of Aquatic Sciences of Chongqing, College of Fisheries, Southwest University, Chongqing 402460, China; 2Key Laboratory of Tropical & Subtropical Fishery Resource Application & Cultivation of Ministry of Agriculture, Pearl River Fisheries Research Institute, Chinese Academy of Fishery Sciences, Guangzhou 510380, China; hongxiaoyou1216@163.com (X.-Y.H.); zhongchy9@mail2.sysu.edu.cn (C.-Y.Z.); xuling92@163.com (X.-L.W.); 15013083966@163.com (X.-P.Z.)

**Keywords:** male germ cells, cryopreservation of genetic resource, cell culture, American shad

## Abstract

**Simple Summary:**

Cryopreservation and in vitro culture of germ cells are key techniques for the genetic resource preservation of the declining population of American shad. Two types of cryopreserved samples, namely testis pieces and testicular cells of American shad, were comparatively analyzed for cell viability. The results showed that the cell viability of the cryopreserved testis pieces was much higher than that of the cryopreserved testicular cells. The viability of the cells from the cryopreserved testis pieces ranged from 65.2 ± 2.2 (%) to 93.8 ± 0.6 (%), whereas the viability of the dissociated cells after cryopreservation was 38.5 ± 0.8 (%) to 87.1 ± 2.6 (%). Moreover, the testicular cells isolated from the post-thaw testicular tissue could be cultured and propagated in vitro. Our findings would benefit further investigations on genetic resource preservation and other manipulations of germ cells in a commercially and ecologically important fish species.

**Abstract:**

Germ cells, as opposed to somatic cells, can transmit heredity information between generations. Cryopreservation and in vitro culture of germ cells are key techniques for genetic resource preservation and cellular engineering breeding. In this study, two types of cryopreserved samples, namely testis pieces and testicular cells of American shad, were comparatively analyzed for cell viability. The results showed that the cell viability of the cryopreserved testis pieces was much higher than that of the cryopreserved testicular cells. The viability of cells from the cryopreserved testis pieces ranged from 65.2 ± 2.2 (%) to 93.8 ± 0.6 (%), whereas the viability of the dissociated cells after cryopreservation was 38.5 ± 0.8 (%) to 87.1 ± 2.6 (%). Intriguingly, the testicular cells from the post-thaw testicular tissue could be cultured in vitro. Likewise, most of the cultured cells exhibited germ cell properties and highly expressed Vasa and PCNA protein. This study is the first attempt to effectively preserve and culture the male germ cells through freezing tissues in the American shad. The findings of this study would benefit further investigations on genetic resource preservation and other manipulations of germ cells in a commercially and ecologically important fish species.

## 1. Introduction

In sexually reproducing organisms, germ cells are the unique cells transmitting the genetic information to the progeny. Among them, both primordial germ cells (PGCs) and spermatogonial stem cells (SSCs) are pluripotent cells that can not only self-renew but also differentiate into gametes [1]. Therefore, studies on germ cell culture and transplantation are attracting increasing attention from biologists. The manipulations techniques related to germ cells have been well-developed in mice, chicks, and some model fish [1,2,3,4,5,6]. However, the advancement of germ cells manipulation techniques in fish is still limited, due to fewer investigations on fish compared to other vertebrates. Beside banking the genetic information of an important species for aquaculture fishes, the efficient cryopreservation of germ cells would be an alternative for securing the samples from rare fish species in wild, which would benefit further investigations and applications of germ cells manipulations in fish [1,5]. So far, cryobanking has been attempted using different types of fish cells [6], including somatic cells [7] and germ cells such as spermatozoa, oocytes, spermatogonia, and PGCs as well as blastomeric embryos [8,9].

To our knowledge, most studies on the cryopreservation of fish have been primarily performed using dissociated cells or cultured cells. Recently, some studies demonstrated the successful cryopreservation of the entire testis of rainbow trout [10,11]. One study reported that the type A spermatogonia (ASGs) in the testis were still viable after a cryopreservation of 728 days [10]. Another study found that the long-term cryopreserved ASGs could proliferate in xenogeneic tissues and could be used for cell transplantation [11]. Since techniques for germ cell cryopreservation are of interest in both genetic resource conservation and breeding, a simple method of simultaneous preserving SSCs and sperm has great practical potential for the genetic resources cryobanking of commercially important and endangered wild fish species. 

American shad, Alosa sapidissima (Wilson, 1811), is an anadromous clupeid fish species natively distributed across the North American coast, from the Gulf of St. Lawrence in Canada to the St. Johns River in Florida. The species has been also introduced into many other places on the North Pacific coast [12,13]. American shad is flavorful fish traditionally caught along with salmon by fisheries industry. Moreover, it is valued as sport fish for anglers, being famous symbol of regional culture. Unfortunately, the population abundance of American shad has been dramatically declining from the early twentieth century, attaining presently the lowest values historically [14], despite the initiation of numerous investigations and extensive protective management measure [15,16]. Therefore, the cryobanking of gametes and germ cells is one available tool for the effective protection of species from extinction. Aiming at developing techniques such as the preservation and restoration of the genetic resources of the endangered wild fish species, we managed to find a way to efficiently identify and manipulate germ cells in American shad. In our previous study, the migrating PGCs and germ cells differentiation during gametogenesis in American shad were primarily investigated through analyzing the expression of germ-line gene *vasa* [17]. In the present report, we performed a comparative analysis of the cell viability between two types of cryopreserved samples, namely testis pieces and dissociated testicular cells. Moreover, we conducted in vitro cell culture to assess potential applications of the post-thaw testicular cells of American shad. The findings of this study will facilitate further investigations on germ cell differentiation and/or manipulations in American shad and other fish species. Moreover, the cryopreservation of fish germ cells, especially the spermatogonial cells, would be an excellent tool for the collection of natural history on fish for the future [18,19,20,21].

## 2. Materials and Methods

### 2.1. Fish and Samples

All the protocols used in this study were approved by the Laboratory Animal Management Committee of Chinese Academy of Fishery Sciences (CAFS), the National Institutes of Health Guide for the Care and Use of Laboratory Animals in China. All fish used in the present study were originated from fish farms located in Guangdong. In total, two groups of male American shad fish were used, which include four 7-month-old fish (weighted 250.0–300.0 g) and eight 24-month-old fish (weighted 628.0–850.0 g). Another group of three, 15-month-old male American shad fish (weighted 550.0–625.0 g) and a group of four adult Asian seabass (*Lates calcarifer*) (four-year-old, weighted 3.5~4.2 kg) (Wang et al., 2016) [22] were also used as control groups or method validation groups. Live fish were immersed in a 0.01% solution of buffered MS-222 until gill movement ceased and there was no response to mechanical stimulation (~10 min). All individuals were rinsed with clean water and the gonads were dissected. The gonad index and developmental stage of dissected testes was briefly examined through weighting and histological examination. For this purpose, small pieces of testicular tissues were placed in 10% neutral buffered formalin for overnight fixation and then embedded in paraffin wax. The embedded wax blocks of testicular tissues were then sliced at 4 µm and placed on microscope slides. The obtained sections were then stained according to hematoxylin and eosin procedures (Harry’s hematoxylin for 2 min and 1% eosin for 30 sec) for light microscopic analysis. 

### 2.2. Cryopreservation of Testicular Tissues and Cells 

After collecting tissues for histological examination, the testes were cut into small pieces for cryopreservation, cell viability, and immunocytochemistry. As shown in the experimental flowchart (Figure 1), the fish testes were dissected and cut into small pieces (edge length of 2~3 mm) for the downstream processes: some of the testis pieces were cryopreserved immediately (5 pieces per tube and 3 tubes per fish); above is two or three. Other pieces of testicular tissue were digested to obtain the isolated testicular cells for cryopreservation (10 tubes per fish) and cell viability analysis of fresh sample. For cell dissociation, two testis pieces per fish were digested with collagenase (LS004196, Worthington, Lakewood, NJ, USA) and trypsin (TLS003703, Worthington, Lakewood, NJ, USA) for 30 min at 28 °C, after which the cell suspension was filtered with a 40 µm cell strainer (BD, 352340, Falcon). The cells were briefly washed twice with PBS and divided into two portions, one for cell viability analysis and the other for cryopreservation. For cryopreservation, the cell pellets were re-suspended with 4 °C pre-cooled cryo-protectant (the final concentration of components: 10% DMSO + 40% FBS + 50% DMEM) and frozen following the previous reports [7,11]. All samples for cryopreservation were gradually cooled from 4 °C to −80 °C at a rate of −1 °C/min cooling using the freezing container (NALGENE^TM^ Cryo 1 °C, cat. No. 5100-0001, Thermo Scientific, Waltham, MA, USA) that was pre-cooled overnight in the fridge at 4 °C. The container with sample tubes was kept at −80 °C for 12 h, then, the sample tubes were transferred to and stored in liquid nitrogen till use. After been kept in liquid nitrogen for 120~350 days, the cryopreserved samples were thawed in a water bath at 37 °C (up to 3 min for tissue pieces and up to 1.5 min for cell suspensions) when needed. 

### 2.3. Immunocytochemistry

Immunostaining was performed on sections of fresh testis tissue, freshly dissociated testicular cells, and post-thaw testicular cells. Fluorescent immunostaining was performed following the previously described procedures [17,23,24]. The cells and sections were visualized using the TSA Plus fluorescence kit (NEL74100KT, PerkinElmer, Rodgau, Germany) with primary antibodies against Vasa (anti-Vasa antibody of gibel carp [24] and PCNA (Sigma, P8825, St. Louis, MO, USA), the secondary antibodies of the horseradish peroxidase (HRP) conjugated anti-rabbit IgG (BA1054; BOSTER, Wuhan, China), and HRP-conjugated anti-mouse IgG (BA1050, BOSTER, Wuhan, China). For counter staining of nuclei, 2 μg/mL propidium iodide (PI, P3566, Life Technologies, Carlsbad, CA, USA) was used as described in our previous study [17]. The stained cells and sections were observed under a confocal microscope (LSM 800, Zeiss, Dublin, CA, USA) and photographed using ZEN 2 imaging software (Zeiss, USA).

### 2.4. Assessment of Cell Viability

Cell viability of both types of cryopreserved samples were assessed and compared to that of fresh testis. For cryopreserved tissues (only for adult testis), the thawed pieces were first squeezed in PBS and the supernatant was centrifuged at 1500 rpm 5 min for sperm collection for downstream analysis. Moreover, the tissue blocks were digested with trypsin to obtain dissociated cells. Testicular cells were washed twice with 1× PBS and suspended in DMEM with 10% FBS. Before the analysis of cell viability, the cell dissociation was briefly checked by staining the testicular cells with Hoechst 33342 (MP Biomedicals, Irvine, CA, USA). It was reported that the Acridine Orange (AO) could stain the live cells, whereasthe propidium iodine (PI) stains the nucleus of dead or intact cells, so that the viability of stained cells could be examined under the Cellometer as described previously [25]. Therefore, in this study, all cells were stained with the ViaStain AO/PI Staining Solution in the PBS Counting Kit (Cat # CS2-0106, Vision CBA, Nexcelom, Lawrence, MA, USA), according to the vendor’s manual, and the cell viability was determined by the Cellometer (Cellometer Vision, Auto2000, K2, Vision; Vision-312-0364, Nexcelom, USA), equipped with the software FCS EXPRESS (DE NOVO SOFTWARE, Solvusoft Corporation, Las Vegas, NV, USA) as previously described [25]. Briefly, the Cellometer and the software could simultaneously monitor the two types of fluorescence, namely fluorescent 1 (AO for live cell) and fluorescent 2 (PI for dead cell), such that cell viability was calculated with the following formula: viability (%) = [number of fluorescent 1 cells/(number of fluorescent 1 cells+ number of fluorescent 2 cells)] × 100. Besides, the software would also provide the sizes of cells examined. Each sample was examined twice and at least three biological samples were examined per group.

### 2.5. In Vitro Culture of the Testicular Cells from Cryopreserved Samples

To further verify the cell viability of the germ cells after cryopreservation, cell culture was conducted as described in our previous study [23]. The thawed testis pieces were dissociated and plated on gelatin-coated six-well plates that were then incubated at 28 ± 0.5 °C. Complete cell medium was used following the previous study [26]. The cultured cells were observed on day 10 and day 30 under an inverted microscope (Axio Observer Z1, Zeiss, Dublin, CA, USA) and photographed using ZEN 2 imaging software.

### 2.6. Statistical Analysis

All calculations were performed with Sigma Stat 3.5 (SYSTAT Software, Inc., San Jose, CA, USA). Data were plotted with Sigma Plot 10.0 (SYSTAT Software, Inc. USA). The data were analyzed using one-way ANOVA with Student T-test. The *p* value < 0.05 was considered statistically significant (*) and *p* value < 0.01 was considered highly statistically significant (**). Values are presented as the means ± SEM (standard error of the mean).

## 3. Results

### 3.1. The Cellular Composition of American Shad Testes

The fish dissection and further experiments shown in Figure 1 were conducted following a strict animal fare rule and standard protocol. The gonad index and developmental stage of all American shad used in this study were examined (Appendix A). The examination of cell dissociation by Hoechst 33342 (MP Biomedicals, USA) staining showed that both the cryopreserved testicular cells and frozen tissues were well dissociated into single cells (Appendix A) before further analyses. As reported in our previous study on American shad, the Vasa protein is mainly and dynamically expressed in germ cells at different stages during gametogenesis [17]. Moreover, PI staining the nuclei of cells could distinguish male germ cells at the early stages, including spermatogonia, spermatocytes from those of late stages, such as spermatids, and sperm based on the size and morphology of the nuclei [24]. In this study, immunostaining with an anti-Vasa antibody and nuclear staining by PI were applied to distinguish the critical stages of germ cells and somatic cells in the examined testicular tissues (Figure 2). In the sections of adult testis, the Vasa signal was strongly detected in the cytoplasm of spermatogonia and spermatocytes, moderately detected and concentrated as particles in spermatids and sperm, but barely detected in somatic cells (Figure 2a–c). Similarly, in the dissociated testicular cells of adult fish, the Vasa protein was highly expressed in the cytoplasm of spermatogonia and spermatocytes, and it was concentrated as dots in spermatids and sperm (Figure 2d–f). Only a small portion of the cells detected did not express Vasa, which should be the somatic cells in testis. Therefore, the Vasa/PI staining showed that the adult testis consists of few somatic cells and a large number of germ cells, including spermatogonia, spermatocytes, spermatids, and spermatozoa (Figure 2c,f). The nuclei size of spermatids, sperm, and some somatic cells is similar and difficult to distinguish from each other through nuclear staining; these cells were classified into one group by the Cellometer during counting. Likewise, using AO/PI staining and cell analysis with the Cellometer, the cellular composition of the testes from America shad at different ages was quantitatively determined as described in the manual of the Cellometer. It was found that the seven-month-old testis was composed of ~26.0% spermatogonia, ~48.7% spermatocytes, and ~19.2% spermatids, and somatic cells, whereas the two-year-old testis was composed of ~13.2% spermatogonia, ~7.6% spermatocytes, ~79.2% spermatids, sperm, and somatic cells (Appendix A).

**Figure 1 biology-11-00790-f001:**
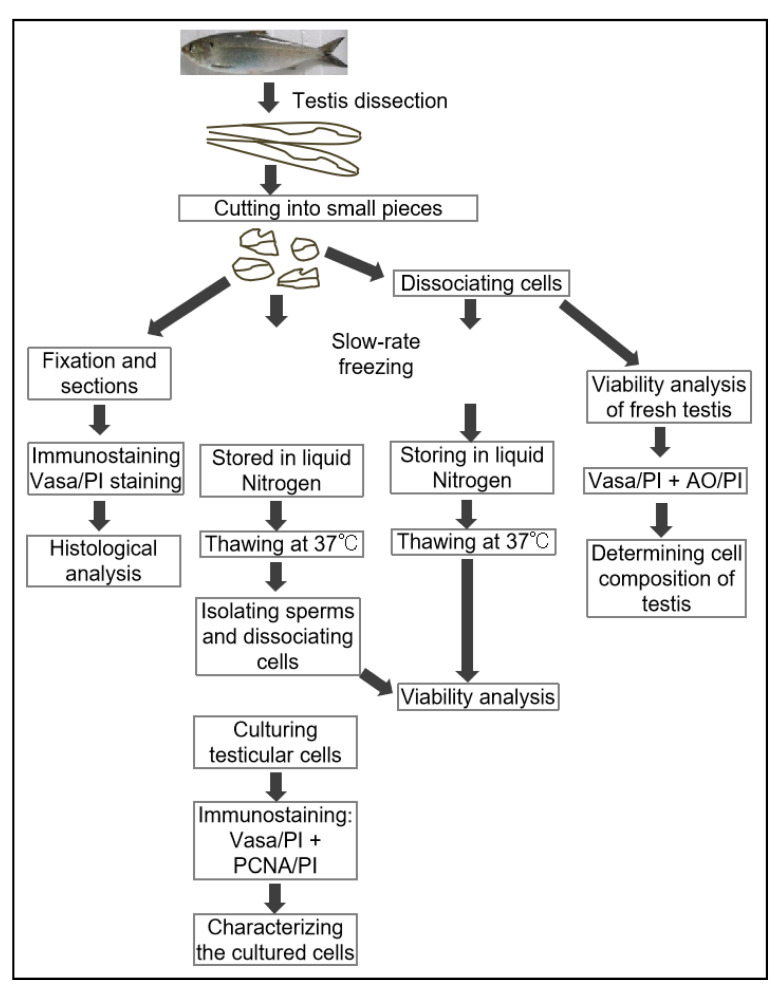
The flowchart of experiments. In total, 19 fish were used in this study, including 4 American shad fish that were 7 months in age (weighted 250.0–300.0 g), and 8 that were 24-months old (weighted 628.0–850.0 g). Moreover, a group of three male American shad fish that were 15-months old (weighted 550.0–625.0 g) and 4 adult Asian seabass (*Lates calcarifer*) (4-year-old, weighted 3.5~4.2 kg) were used as control groups or method validation. Live fish were immersed in a 0.01% solution of buffered MS-222 until gill movement ceased and there was no response to mechanical stimulation (~10 min). All individuals were rinsed with clean water and the gonads were dissected. All fish testes were dissected and cut into small pieces (edge length of 2~3 mm) for the downstream processes: some of the testis pieces were cryopreserved immediately (5 pieces per tube and 3 tubes per fish); some of pieces were digested to obtain isolated testicular cells for cryopreservation (10 tubes per fish) and cell viability analysis of fresh samples. Two tubes of samples from one fish were analyzed in technical duplicates, and at least three fish were examined for each group or batch. The histological analysis (H&E staining and immunostaining) were conducted in three samples (of fish) per batch. All data are shown in percentages (%) and the mean ± SD in tables, *n* = 6.

### 3.2. Cryopreservation of Testicular Cells and Tissues

After freezing/thawing, the testicular cells and tissue pieces were analyzed for cell composition and viability. The results showed that there was no difference in the proportions of cells at different stages between the cryopreserved tissues and isolated cells (Appendix A). The viability of the cells isolated from fresh testes were very high, up to 93~98% (Table 1 and Table 2). As for the cryopreserved testicular cells, the viability of spermatogonia, spermatocytes, and spermatids with somatic cells were 38.5–39.2%, 33.8–37.2% and 46.5–49.4%, respectively (Table 1 and Table 2). In the dissociated cells from cryopreserved tissue pieces, the viability of spermatogonia, spermatocytes, spermatids, and other cells was approximately 51.6–65.2%, 58.7–62.9%, and 59.2–62.5%, respectively (Table 1 and Table 2). There was a statistically significant difference in the viability of spermatogonia and spermatocytes between these two methods of cryopreservation (*p* < 0.05). The spermatogonia and spermatocytes from the frozen tissue blocks showed significantly higher viability than those from the cryopreserved cells. Moreover, the recorded results indicate that bigger cells, i.e., spermatogonia and spermatocytes, were less viable when compared to smaller cells, i.e., spermatids and sperm, with the exception of the cryopreserved tissue blocks from 2-year old fish examined in this study (Table 1 and Table 2).

### 3.3. Validation of the Cryopreservation Method with Another Batch of Samples

To briefly demonstrate the effects of cryopreservation time on cell viability and validate whether the cryopreservation of testis pieces is robust and suitable for other fish species, another two batches of cryopreserved tissues were evaluated, including the American shad testis pieces (15-month-old), which were cryopreserved over 120 days, and the Asian seabass testis pieces (~4-year-old), which were cryopreserved over 300 days. The results showed that, in American shad testis frozen for ~120 days, the viability of the spermatogonia, spermatocytes, spermatids and somatic cells was around ~72.1%, 80.7% and ~76.1%, respectively, which is higher than those of cells from the testis cryopreserved for 300 days (Table 1, Table 2 and Table 3). Similarly, in the Asian seabass, the viability of spermatogonia, spermatocytes, spermatids, and somatic cells was 35.5%, 56.4% and~68%, respectively (Table 3). These also confirmed that the cryopreserved smaller cells, i.e., spermatids and sperm, exhibited higher viability (Appendix A). 

### 3.4. Cell Proliferation of Cryopreserved Tissue

To further evaluate the ability of post-thaw cells to proliferate, the testis pieces of American shad were briefly digested with enzyme and plated onto a six-well plate for in vitro culture. On day 10 and day 30, the cultured cells were observed, and most of cells growing out from the tissue blocks were epithelial-like in the primary culture (Figure 3a). Additionally, the cells slowly proliferated in the first passages and could be passaged at intervals of 5 or 6 days at a ratio of 1:2. After 30 days or the 6th passage, cells in a monolayer were obtained (Figure 3b). Most of the cultured cells were large in size and possessed prominent, round, and large nuclei (Figure 3b). To further characterize the cultured cells, the expression of Vasa (Figure 4a–c) and PCNA (Figure 4d–f) were also examined. Both the germ cell-specific protein Vasa and cell proliferation marker PCNA were abundantly detected in most of the cultured cells. In particular, the Vasa signals were distributed in the perinuclear cytoplasm of cultured cells (Figure 4a–c), whereas PCNA was dominantly detected in the nuclei of cells (Figure 4d–f). 

## 4. Discussion

The cryobanking of germ cells has important applications in reproductive practices in marine or freshwater aquatic fish species through simplifying the broodstock management [5,6,10,20]. It has also been applied in maintaining important strains of laboratory fish species [8,27,28]. Likewise, the cell viability is usually the main concern during the cryopreservation of cells or tissues. Among the staining methods for cell viability analysis, trypan blue staining is a common and simple method to analyze cell viability [29]. However, researchers have found that trypan blue could not stain all dead cells, which led to false-negative errors for cell viability analysis [30]. After AO/PI staining, germ cells at different stages became visible, the living and dead cells could be easily distinguished [25]. Therefore, in the present study, the co-staining of AO/PI was adopted to determine cell viability, so that alive and dead cells (or intact cells) can be easily and rapidly distinguished from each other.

Previous studies have documented that the viability of cryopreserved germ cells is determined by a variety of factors, including sample treatment, the components of cryo-protectant, cooling and recovery rates, freezing temperature, and storage time [31]. The improved outcomes of this study may be attributed to the use of DMSO in the cryoprotectant, which has been well studied in other fish species; for instance, the viability of cryopreserved sperm was 85.25% in *N. albiflora* [32] and 92.91% in *Sparus microcephalus* [33]. In particular, 10% DMSO could improve the viability, mobility, and life-span of frozen-thawed sperm to the levels that are comparable to those observed in fresh sperm [34]. Thus, in this study, a cryoprotectant containing 10% DMSO was adopted to efficiently cryopreserve the testicular cells in fish, including American shad and Asian seabass. Moreover, we found that this cryoprotectant could not always provide a high viability for male germ cells at the early stages, like that of frozen spermatids or sperm. Therefore, more extensive investigations are needed to optimize the conditions to cryopreserve the germ cells at different stages in fish. Likewise, the viability of American shad cells cryopreserved for ~300 days is much lower than that of cells after a 120-day cryopreservation, so maybe the storage time also affects the cell viability of frozen testis in American shad, but to well address this, further investigations are needed in the future. 

Conventionally, the cryopreservation was carried out with isolated or cultured cells, which could be applied in further studies after thaw/recovery. This study revealed that the viability of germ cells, especially for spermatogonia and spermatocytes in cryopreserved tissue, could be significantly higher than in cryopreserved isolated cells. This finding was similar to a study in Siberian sturgeon, which showed that the number of dead cells from whole tissue cryopreservation was smaller than that found when using other methods [11]. To sum up, in this study, the cryopreservation of tissue pieces preserved most of the germ cells at different stages, which is more efficient than the traditional methods using dissociated cells such as for SSCs [19] or sperm preservation [35].

The opportunity of cryobanks creation for viable gametes, pluripotent cells (primordial germ cells, spermatogonia, oogonia, etc.), and somatic tissues represents a valuable tool in biodiversity conservation and in aquaculture production through preserving the genetic resources of valuable breeds and endangered fish species [5,6,7,9,19,27,36,37]. Cryopreserved PGCs, SGs, and OGs can be transplanted into heterogenic recipient fish species and possess abilities to migrate to the recipient gonads, as well as begin to induce the gametogenesis of functional heterologous gametes in their gonads [5,11,31]. Thus the transplantation of cryopreserved germ cells can be another powerful tool for the preservation and recovery of fish genetic resources [5,6,31]. In this context, the cryopreserved tissues in this study might be used as donor cells for fish interspecies cell transplantation in the future.

Another potential application of cryopreserved testis pieces concerns fish regeneration, where a sufficient number of cells can be recovered from a few milligrams of tissue. Methods to obtain cultured cells from fish tissues have been developed for many species [26,38,39,40], but the quality and growth capacity of the cultured cells vary under different conditions [41]. Therefore, the culture conditions should be optimized and this could be performed separately from the sampling and cryopreservation process. Here, in our study, the cryopreserved gonad cells were in vitro cultured and passaged. After several passages, there were more round cells with prominent and large nuclei, which have been proven to be characteristics of germ cells [23,26,42]. Moreover, in most of these cultured cells, the germ cell marker Vasa protein and cell proliferation marker PCNA were highly expressed. It has been well demonstrated that PCNA expression is related to cells’ self-renewal and/or mitotic activity, thus, the high level of PCNA signals detected indicates that most of the cultured cells had a high potency of proliferation or mitotic activity. This implied that the cryopreserved cells were still able to proliferate and could be subjected to in vitro cell culture. Another challenge associated with developing cryopreservation protocols for fish germ cells (GCs) is to develop a successful protocol for the in vitro maturation of GCs after cryopreservation. Such a protocol has been successfully developed for the in vitro maturation of GCs in medaka fish [26]. Taken together, the cryopreserved tissues could efficiently preserve all stages of male germ cells, including spermatogonia, and the germ cells could be propagated through cell culture for downstream cells manipulations, such as direct differentiation or maturation. 

## 5. Conclusions

In summary, testicular cells of American shad were isolated and identified through immunofluorescence staining with an anti-Vasa antibody and PI. Two types of cryopreserved samples were evaluated, namely the cryopreserved tissue pieces and the cryopreserved dissociated cells. The viability of post-thaw cells was comparatively analyzed between the two types of samples, and it was found that the cryopreserved testis pieces provided higher viability of testicular cells. Importantly, the cells from cryopreserved tissues were able to proliferate and could be used for in vitro cell culture. Thus, the technique of efficient cryopreservation developed in this study would be a powerful tool for the preservation of the genetic resources of endangered and valuable fish stocks for aquaculture. Yet, more extensive investigations are still needed to optimize the protocols for cryopreserving larger germ cells in fish. 

## Figures and Tables

**Figure 2 biology-11-00790-f002:**
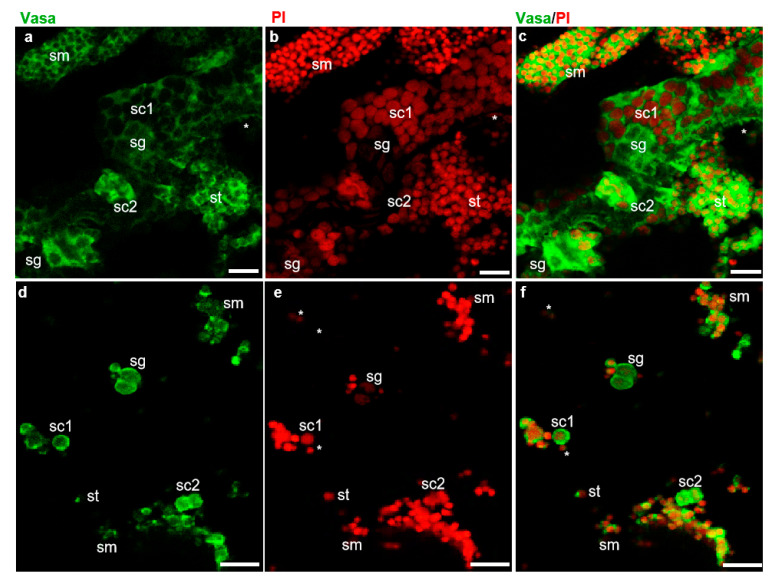
Identification of testicular cells by fluorescent immunostaining. Fluorescent immunostaining on testis sections and isolated testicular cells of adult American shad (2Y^+^). (**a**–**c**), sections. (**d**–**f**), Testicular cells. Green fluorescence is for Vasa protein (**a**,**c**) and red fluorescence is the nuclear staining with propidium iodide (PI) (**b**,**e**); (**c**) and (**f**) are the merged images. sg, spermatogonia. sc, spermatocytes. st, spermatids. sm, spermatozoa and somatic cells indicated with asterisks. Scale bars, 20 µm.

**Figure 3 biology-11-00790-f003:**
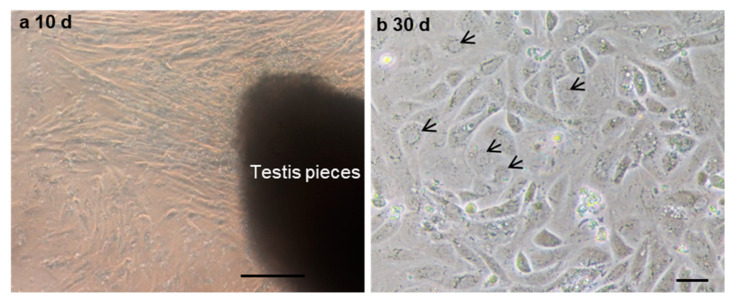
The in vitro culture of cryopreserved testicular cells. The testicular cells were cultured at day 10 (**a**) and day 30 (**b**). Arrows indicate the nuclei of cells. Scale bars, (**a**) 100; (**b**) 25 µm.

**Figure 4 biology-11-00790-f004:**
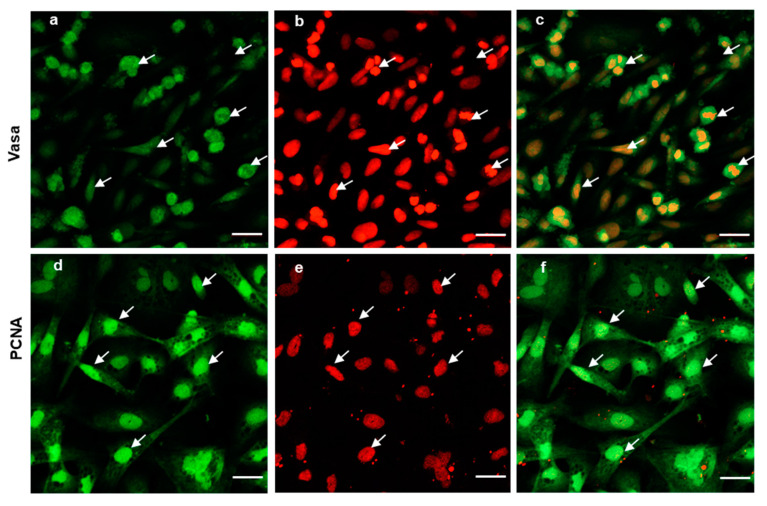
Expression of Vasa and PCNA in the cultured cells. Fluorescent immunostaining analysis of the cultured cells at 6th passage (cultured over 30 days); the cells were stained with polyclonal anti-Vasa antibody (green in **a**), and Monoclonal anti-PCNA antibody (green in **d**). Nuclei were stained with PI (red in **b**,**e**). Merged images (**c**,**f**). The arrows indicate the cells expressing both Vasa and PCNA. Scale bars, 20 µm.

**Table 1 biology-11-00790-t001:** The viability of testicular cells from the fresh and cryopreserved testes from 7-month-old shad.

Cell Types	Cell Viability (%)			*p*-Value
	Dissociated Cells from Fresh Testis	Cryopreserved Dissociated Cells	Cryopreserved Tissue Blocks	
Spermatogonia	93.4 ± 5.2	39.2 ± 4.6	51.6± 1.3	0.023
Spermatocytes	96.7 ± 2.3	33.8 ± 4.2	62.9 ± 2.2	0.001
Spermatids and others	96.3 ± 1.8	46.5 ± 1.2	62.5 ± 2.4	0.008

The dissociated testicular cells and tissue blocks were frozen and stored in liquid nitrogen over 300 days and thawed for analysis. The data are shown in percentages (%) and the mean ± SEM, *n* = 12 (3 fish × 2 tubes of samples × 2 counts). The *p*-value is for the significance of difference in cell viability between cryopreserved dissociated cells and frozen tissue blocks.

**Table 2 biology-11-00790-t002:** The viability of testicular cells from fresh and cryopreserved testis from 2-year-old shad.

Cell Types	Cell Viability (%)			*p*-Value
	Dissociated Cells from Fresh Testis	Cryopreserved Dissociated Cells	Cryopreserved Tissue Blocks	
Spermatogonia	97.2 ± 2.2	38.5 ± 0.8	65.2± 2.2	0.003
Spermatocytes	98.7 ± 0.6	37.2 ± 4.4	58.7 ± 2.6	0.034
Spermatids and others	96.1 ± 3.1	49.4 ± 0.8	59.2 ± 1.2	0.001

The dissociated testicular cells and tissue blocks were frozen and stored in liquid nitrogen over 300 days and thawed for analysis**.** The data are shown in percentages (%) and the mean ± SEM, *n* = 12 (3 fish × 2 tubes of samples × 2 counts). The *p*-value is for the significance of difference in cell viability between the cryopreserved dissociated cells and frozen tissue blocks.

**Table 3 biology-11-00790-t003:** The method of cryopreservation validated in new batch of American shad and Asian seabass.

Cell Types	Cell Viability (%)	
	As Testis	Sb Testis
Spermatogonia	72.1 ± 3.9	35.5 ± 8.6
Spermatocytes	80.7 ± 2.2	56.4 ± 11.5
Spermatids and others	76.1 ± 3.2	68.0 ± 0.4

As testis: testis pieces of 1-year-old American shad after a cryopreservation of 120 days; Sb testis, testis pieces of 4-year-old Asian seabass after a cryopreservation of ~300 days. The data are shown in percentage (%) and the mean ± SEM, *n* = 12 (3 fish × 2 tubes of samples × 2 counts).

## Data Availability

Not applicable.

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
