# Peer review of "Restoring Genetic Resource through In Vitro Culturing Testicular Cells from the Cryo-Preserved Tissue of the American Shad (Alosa sapidissima)"

_biology, 2022, doi:10.3390/biology11050790_

Round 1

Reviewer 1 Report

The manuscript provides some interesting information, however the data and experimental design could be improved.

  1. The results of cell viability are key data, and the authors said that co-staining of AO/PI was adopted to determine cell viability, so that alive and dead cells (or intact cells) can be easily and rapidly distinguished from each other (page 9, line 297-298). Therefore, the pictures of AO/PI staining should be revealed but not only counting numbers in table.
  2. The authors motioned that the viability of spermatogonia and spermatocytes were lower than that of smaller cells, such as spermatids and sperm, in any samples examined in this study (page 6, line 230-231), however the data of cryopreserved tissue block did not show this in table1 and table2. The statistical analysis between different types of cells is required.
  3. Since cell viability of the cryopreserved testis pieces was much higher than that of the cryopreserved testicular cells, how about the physiological functions of these spermatids and sperms? Compare the outcomes of in vitro fertilization in two groups maybe necessary.
  4. Please rearrange the tables because the present typography is difficult to read.
  5. The purpose of table 3 is unclear and the conclusion is missing. They evaluated another two batches of cryopreserved tissues, but the all of the conditions are nothing in common, such as the animal model (American shad vs. Asian seabass); the age of fish (1 year vs. 4 years), and the time of cryopreservation (120 days vs. ~300 days). I don’t understand why they want to compare these two groups, and cannot get any information in this data. 

Author Response

The manuscript provides some interesting information, however the data and experimental design could be improved.

  1. The results of cell viability are key data, and the authors saidthat co-staining of AO/PI was adopted to determine cell viability, so that alive and dead cells (or intact cells) can be easily and rapidly distinguished from each other (page 9, line 297-298). Therefore, the pictures of AO/PI staining should be revealed but not only counting numbers in table.

Response: Good suggestions. In this study, we counted the cells using a cell counter (Cellometer Vision, Auto2000, K2, Vision; Vision-312-0364) which were proved to be more sensitive and accurate basing on the cells staining with AO/PI kit  (Cat # CS2-0106, Vision CBA) (Chan et al., 2017)(details shown in Material and method section). Therefore, here we didn’t confirm this by taking pictures with high resolution. By the way, the pictures taken by the counter were added as the supplemental files for reviewer to refer(fig.S5).

  1. The authors motioned that the viability of spermatogonia and spermatocytes were lower than that of smaller cells, such as spermatids and sperm, in any samples examined in this study (page 6, line 230-231), however the data of cryopreserved tissue block did not show this in table1 and table2. The statistical analysis between different types of cells is required.

Response: We modified the related description as “Moreover, the recorded results indicate that bigger cells, i.e. spermatogonia and spermatocytes were less viable when compared to smaller cells, i.e. spermatids and sperm, with the exception to cryopreserved tissue blocks from 2-year old fish......” in line 244-248.

  1. Since cell viability of the cryopreserved testis pieces was much higher than that of the cryopreserved testicular cells, how about the physiological functions of these spermatids and sperms? Compare the outcomes ofin vitro fertilization in two groups maybe necessary.

Response: Good suggestions. In this study, we focused on culturing the cells from the cryopreserved tissues, in a word, we were mainly interested in finding the efficient ways to cryopreserve and recover the genetic resource, and maybe we would determine the functions of spermatids and sperms in the future study, especially in the studies on sperm cryopreserving since these kind of investigations had been conducted in many species, including American shad (Wang M H, 2015. In Chinese).

  1. Please rearrange the tables because the present typography is difficult to read.

Response: Thanks a lot for valuable comments. We modified the tables. 

  1. The purpose of table 3 is unclear and the conclusion is missing. They evaluated another two batches of cryopreserved tissues, but the all of the conditions are nothing in common, such as the animal model (American shad vs. Asian seabass); the age of fish (1 year vs. 4 years), and the time of cryopreservation (120 days vs. ~300 days). I don’t understand why they want to compare these two groups, and cannot get any information in this data. 

Response: Valuable comments. The experiment of table3 is just for a control or validation of the method for tissue preservation using different group of samples, and even different species, which proves that tissue preservation applied here could be considered as a robust method for preserving the genetic resource of fish, and the male seabass always matures at the age of 3-4 years, therefore we collected the testis from 4-Y seabass. Moreover,the new group of Asian seabass at 1year old was used to determine the effects of preserving time (120d vs 300d) on cell viability. We have added “Moreover, the recorded results indicate that bigger cells, i.e. spermatogonia and spermatocytes were less viable when compared to smaller cells, i.e. spermatids and sperm, with the exception to cryopreserved tissue blocks from 2-year old fish” in line 244-248.

Reviewer 2 Report

Dear Editors,

Dear Authors,

The reviewed manuscript entitled: “Restoring genetic resource through in vitro culturing testicular cells from the cryo-preserved tissue of the American Shad (Alosa sapidissima)” represents valuable and very interesting insight to the biotechnology and genome engineering. The performed under the reviewed manuscript research provides important information that can be used in the future genetic resource preservation of fishery important and simultaneously endangered fish species, which is the American Shad. Moreover, the described, in the reviewed study, protocol for testicular cells cryopreservation can be also used for other fish species. Despite high value of reviewed the manuscript, it requires serious improvements starting from information clarification and ending to the data presentation and language quality. It would take a lot of work and space to list all questions, fixes, and suggestions here so all of them has been provided in the attached pdf file.

In conclusion, I recommend re-evaluating the manuscript for possible publication in the Biology Periodical after major revision with accordance to the remarks placed in the attached pdf file.

Thank you for another interesting manuscript that I could review!

Author Response

Dear Professor,

We really appreciate you for your patience and efforts on our manuscript. Your valuable comments and suggestions greatly helped us with improving the manuscript writing. We responded and revised the points one by one (red in revised manuscript). Also, we added more supplemental files to support the conclusion of this paper.

Thank you very much!

Best regards!

Hongyan

Reviewer 3 Report

In this MS, Xu et al. perform a comparative analysis on the cell viability between 2 types of cryopreserved samples, namely testis pieces and dissociated testicular cells from the American shad (a threatened fish species). The authors hope that the findings of this study will enable further investigations on the reproductive biology of this species, and thereby help with genetic manipulations and captivity breeding of the Shad and other endangered anadromous fish. Moreover, the cryopreservation technique for fish germ cells (and their subsequent revival) hold promise in the field of reproductive biology at large. Cryobanking of germ cells has great potential and medical applications for all species, especially given that we may be facing the next mass extinction event of our biodiversity. Overall, I think this MS adds important information to the field and can be considered for publication following minor revisions. The authors will benefit from a good proof-read of their MS, and/or using more scientifically strong language.

Comments:

  1. Line14: what do the authors mean by the term “cellular engineering breeding”? Do you mean in vitro fertilization?
  2. Line20: what does “proved to be propagated” mean? It can be replaced by the word revive, if this is the meaning the authors intend to make
  3. Line28-29: “In sexually reproducing organisms, germ cells are the unique cells transmitting the genetic information to the progeny” please make this change
  4. Line37: delete “field species”. Sentence can read …difficulty in obtaining fresh samples from the wild”
  5. Line132: is it freshly dissected testis? Either add “dissected” or change freshly to fresh
  6. Fig1: This is a very useful schematic. Is it possible to add further details in this fig? For example, can the exact number of tissues used for each procedure be also added (perhaps next to the arrows?). Also is the paragraph line 197-209 the figure legend? This needs to be formatted correctly so that it is not confused as main text. Same comment applied for figure legends for all subsequent figures.
  7. Line172: I am not sure about the authors claim that Vasa protein is “specifically” expressed in germ cells. Yes Vasa is expressed by germ cells, but it is also expressed by a bunch of other cells (especially stem cells). I would caution the authors from using the word “specifically” unless they actually know this to be true and can reference their claim.
  8. Line174: “based” on the size and morphology of nuclei
  9. Fig2: The writing “Vasa”, “PI” and “Vasa/PI” is obscured by the immunostained image. Please change placement of text. Also, the images are so over-saturated that it is impossible to tell apart the differences in sub-cellular localization as the authors describe in Line177-186. Are higher magnification images for the same available? They could be added to the figure as a zoomed insert.
  10. Q. Do the testes of the American Shad produce germ cells and mature gametes life long? Also has a similar experiment been done to try and cryopreserve the American Shad Ova? Finally, have primordial germ cells, or stem cells been induced to produce mature gametes in vitro for this species?
  11. Table1: This table needs to be reformatted. In the form it shows up, labels are moved. What does the numbering (8-22) refer to in this table? Please remove it if it is unnecessary
  12. With regards to the expression of Vasa and PCNA as the only qualifiers to bestow germ cell fate to the revived cryopreserved cells, can any other method be used to ascertain that the revived cells have not reverted to a pluripotent fate (which would also express Vasa and PCNA)? Do the cells, for example, go on to become mature gametes upon prolonged culture? The authors should provide additional evidence to support their claim that they can gt back functional gametes at the end of cryopreservation, and/or be able to induce the pluripotent germ-cell like cells to becomes gametes.
  13. Along the same lines, can these cultured cells be used for in vitro fertilization in this species? Can the authors comment on this, and also on the status of cryopreservation of the female Shad ova?

Author Response

In this MS, Xu et al. perform a comparative analysis on the cell viability between 2 types of cryopreserved samples, namely testis pieces and dissociated testicular cells from the American shad (a threatened fish species). The authors hope that the findings of this study will enable further investigations on the reproductive biology of this species, and thereby help with genetic manipulations and captivity breeding of the Shad and other endangered anadromous fish. Moreover, the cryopreservation technique for fish germ cells (and their subsequent revival) hold promise in the field of reproductive biology at large. Cryobanking of germ cells has great potential and medical applications for all species, especially given that we may be facing the next mass extinction event of our biodiversity. Overall, I think this MS adds important information to the field and can be considered for publication following minor revisions. The authors will benefit from a good proof-read of their MS, and/or using more scientifically strong language.

Response: we are appreciated for the reviewer’s efforts and patience on our manuscript. And the authors have read through the manuscripts and tried to improve the writing. 

Comments:

  1. Line14: what do the authors mean by the term “cellular engineering breeding”? Do you mean in vitro fertilization?

   Response: “cellular engineering breeding” means cell transplantation.

  1. Line20: what does “proved to be propagated” mean? It can be replaced by the word revive, if this is the meaning the authors intend to make

Response: we changed it to be “the testicular cells from the post-thaw testicular tissue could be cultured in vitro.”

  1. Line28-29: “In sexually reproducing organisms, germ cells are the unique cells transmitting the genetic information to the progeny” please make this change

Response: Thanks a lot for the valuable suggestion. We changed this sentence as required.

  1. Line37: delete “field species”. Sentence can read …difficulty in obtaining fresh samples from the wild”

Response: Deleted already.

  1. Line132: is it freshly dissected testis? Either add “dissected” or change freshly to fresh

Response: Changed ‘freshly’ into ‘fresh’.

  1. Fig1: This is a very useful schematic. Is it possible to add further details in this fig? For example, can the exact number of tissues used for each procedure be also added (perhaps next to the arrows?). Also is the paragraph line 197-209 the figure legend? This needs to be formatted correctly so that it is not confused as main text. Same comment applied for figure legends for all subsequent figures.

Response: We formatted the figure legend. In this study, we used several groups of fish for experiment and the details on samples were described in the legend and the text.

  1. Line172: I am not sure about the authors claim that Vasa protein is “specifically” expressed in germ cells. Yes Vasa is expressed by germ cells, but it is also expressed by a bunch of other cells (especially stem cells). I would caution the authors from using the word “specifically” unless they actually know this to be true and can reference their claim.

Response: Good suggestion. We replaced ‘specifically’ with ‘mainly’.

  1. Line174: “based” on the size and morphology of nuclei

Response: Changed it already.

  1. Fig2: The writing “Vasa”, “PI” and “Vasa/PI” is obscured by the immunostained image. Please change placement of text. Also, the images are so over-saturated that it is impossible to tell apart the differences in sub-cellular localization as the authors describe in Line177-186. Are higher magnification images for the same available?

They could be added to the figure as a zoomed insert.

Response: we modified the labeling of Fig.2 . We had used the figures with highest magnification in which all the germ cells at different developmental stages and somatic cells could be easily observed as shown in the figure 2.

  1. Do the testes of the American Shad produce germ cells and mature gametes life long? Also has a similar experiment been done to try and cryopreserve the American Shad Ova? Finally, have primordial germ cells, or stem cells been induced to produce mature gametes in vitro for this species?

Response: Like other fish, the American shad can produce germ cells and mature gametes during a certain period of its life cycle, but not life long. We are trying to cryopreserve the American Shad Ova in our further studies.

  1. Table1: This table needs to be reformatted. In the form it shows up, labels are moved. What does the numbering (8-22) refer to in this table? Please remove it if it is unnecessary

Response: the labels were removed.

  1. With regards to the expression of Vasa and PCNA as the only qualifiers to bestow germ cell fate to the revived cryopreserved cells, can any other method be used to ascertain that the revived cells have not reverted to a pluripotent fate (which would also express Vasa and PCNA)? Do the cells, for example, go on to become mature gametes upon prolonged culture? The authors should provide additional evidence to support their claim that they can gt back functional gametes at the end of cryopreservation, and/or be able to induce the pluripotent germ-cell like cells to becomes gametes.

Response: It’s a great idea. Here, we just want to determine if most of the cryopreserved cells are germ cells (expression Vasa protein) and can proliferate (expressing PCNA protein) under culture condition. And we will analyze the characteristics of the cryopreserved cell in our further studies in the future.

  1. Along the same lines, can these cultured cells be used for in vitrofertilization in this species? Can the authors comment on this, and also on the status of cryopreservation of the female Shad ova?

Response: Good suggestions. Actually, it had been proved that the cultured cells derived from fish testes could be induced and differentiated into motile sperm in our previous study (Hong YH et.al. 2004). However, the amount of the sperm generated from cultured cells is so few and hard to used for fertilization. Also, we are trying to cryopreserve the fish ovarian cells in our further studies.

Round 2

Reviewer 1 Report

The picture of Fig. S4. is vary bad and can not distinguish the  alive and dead cells (or intact cells) cells.

Author Response

The bright and fluorescent emerged pictures were briefly taken and provided by the Cellometer, which is not professional microscopy, therefore the pictures would not have high resolution and quality as those by Microscopy, the fig.S4 was just provide a reference for us.

The PI can only stain the dead or intact cells; the AO can stain the live cells; Hochest (blue) stain all of cells.

Reviewer 2 Report

Dear Editor,

Dear Authors,

The reviewed manuscript has been significantly improved but still some issues must be corrected and clarified. Most of them concerns the results presentation and methods application. Still manuscript require some major/minor revisions. All fixes, questions and comments are included in the attached file.

After corrections the manuscript should be once more evaluated.

Best regards

Author Response

The  pictures in  Fig. S4 were briefly taken and provided by the Cellometer, which is not professional microscopy, therefore the pictures should not have high resolution and quality as those by Microscopy. This figures were just provided for a reference.